# The Phenotype of Celiac Disease Has Low Concordance between Siblings, Despite a Similar Distribution of HLA Haplotypes

**DOI:** 10.3390/nu11020479

**Published:** 2019-02-25

**Authors:** Saana Kauma, Katri Kaukinen, Heini Huhtala, Laura Kivelä, Henna Pekki, Teea Salmi, Päivi Saavalainen, Katri Lindfors, Kalle Kurppa

**Affiliations:** 1Celiac Disease Research Centre, Faculty of Medicine and Life Sciences, Tampere University, 33520 Tampere, Finland; saana.kauma@tuni.fi (S.K.); katri.kaukinen@tuni.fi (K.K.); henna.pekki@tuni.fi (H.P.); teea.salmi@tuni.fi (T.S.); katri.lindfors@tuni.fi (K.L.); 2Department of Internal Medicine, Tampere University Hospital, 33521 Tampere, Finland; 3Faculty of Social Sciences, Tampere University, 33520 Tampere, Finland; heini.huhtala@tuni.fi; 4Tampere Centre for Child Health Research, Tampere University and Tampere University Hospital, 33521 Tampere, Finland; laura.kivela@fimnet.fi; 5Department of Dermatology, Tampere University Hospital, 33521 Tampere, Finland; 6Research Program Unit, Immunobiology, and Department of Medical and Clinical Genetics, University of Helsinki, 00014 Helsinki, Finland; paivi.saavalainen@helsinki.fi

**Keywords:** celiac disease, sibling, phenotype, gluten-free diet, environmental factors, genotype

## Abstract

The factors determining the presentation of celiac disease are unclear. We investigated the phenotypic concordance and the distribution of human leukocyte antigen (HLA) risk haplotypes in affected siblings. One hundred sibling pairs were included. Clinical and histological parameters and HLA haplotypes were compared between the first diagnosed indexes and their siblings. The phenotype was categorized into gastrointestinal, extra-intestinal, malabsorption/anemia, and asymptomatic. The phenotype was fully concordant in 21 pairs. The most common concordant phenotype was gastrointestinal (14 pairs). Indexes had more anemia/malabsorption and extra-intestinal symptoms than siblings (45% vs. 20%, *p* < 0.001 and 33% vs. 12%, *p* < 0.001, respectively). Twenty siblings and none of the indexes were asymptomatic. The indexes were more often women (81% vs. 63%, *p* = 0.008). They were also more often seronegative (11% vs. 0%, *p* = 0.03) and younger (37 vs. 43 year, *p* < 0.001), and had more severe histopathology (total/subtotal atrophy 79% vs. 58%, *p* = 0.047) at diagnosis. The indexes and siblings were comparable in other disease features. Pairs with discordant presentation had similar HLA haplotypes more often than the concordant pairs. The phenotype was observed to vary markedly between siblings, with the indexes generally having a more severe presentation. HLA did not explain the differences, suggesting that non-HLA genes and environmental factors play significant roles.

## 1. Introduction

Celiac disease is an immune-mediated condition with an estimated prevalence of 1–2% in Western countries [1,2,3]. The first-degree relatives of patients have approximately 2–10 times the average risk for the disease, whereas in identical twins the concordance rate can be as high as 80% [4,5,6]. Human leukocyte antigen (HLA) DQ2 and DQ8 haplotypes have been identified as the main genetic risk factors, without which celiac disease is very unlikely [7]. At population level, approximately 40% of individuals have these risk haplotypes, but only a fraction of them will eventually develop the disease [2,7]. This might be partly explained by the effect of non-HLA genes, but considerable differences in the prevalence between genetically similar populations and the rising true incidence support the additional role of environmental factors [3,8,9].

In recent decades, we have also come to appreciate that the phenotype of celiac disease is very heterogeneous. The classical presentation with malabsorption and failure to thrive in early childhood has become rare. Nowadays, patients often have different gastrointestinal or extra-intestinal symptoms that may appear at any age, or they can even be completely asymptomatic [10,11,12]. The reason for this phenotypic diversity remains obscure [13,14,15,16,17], but the observed variability even between identical twins suggests that it is not solely determined by genetics [5,18]. Overall, the concordance of the clinical picture between affected relatives has been scarcely studied. This information could improve our understanding of the complex interactions between genetic and environmental factors in celiac disease, and possibly increase the diagnostic yield of this markedly under-recognized condition [3].

In this study, we aimed to evaluate the concordance of the clinical and histological presentation and the HLA risk haplotypes of untreated celiac disease in close relatives who both have the disease. Specifically, the comparisons were made between the first affected index patients and their siblings, who usually have/had a shared environment in childhood and are genetically markedly similar [6].

## 2. Materials and Methods 

### 2.1. Patients and Study Design

The study was carried out in Finland at the University of Tampere and Tampere University Hospital. Previously diagnosed celiac disease patients and their relatives were invited to participate through advertisements in newspapers and via local celiac societies. All participants were interviewed by a study nurse or physician with expertise in celiac disease, and blood samples were drawn for further serological and genetic analyses between 2006–2010. Relatives with new celiac autoantibody positivity were referred to the local hospital for diagnostic endoscopy. In addition to the interview, the medical records of the patients were surveyed to confirm the diagnosis and to supplement the clinical, histological, and serological data at diagnosis. Diagnosis had to be based on the demonstration of villous atrophy in duodenal biopsy in both children and adults. The exclusion criteria were study refusal and unclear celiac disease diagnosis.

Altogether, 1035 patients and 3031 of their relatives from 732 families were enrolled (Figure 1). Among the 3031 relatives, 148 new cases of celiac disease were detected by screening. Thus, 1183 subjects had either previously diagnosed or newly diagnosed celiac disease. Of this number, 263 were excluded due to insufficient data for the study analyses. Of the remaining 920 patients, only families with at least two affected subjects (*n* = 492) entered the next stage. In order to simplify the statistical evaluation, only two first-affected siblings from families with multiple cases were enrolled. The final study group comprised 200 subjects (100 sibling pairs) who underwent comparison for all study variables as described below (Figure 1). The first diagnosed subject is defined as the index and the later diagnosed subject is defined as the sibling. The 200 patients included in the final analyses were diagnosed between the years 1972–2009.

### 2.2. Clinical Characteristics

The clinical information collected included demographic data and the family history of celiac disease, the main disease presentation/reason for disease suspicion, and the possible presence of co-existing autoimmune conditions, fractures, and malignancies. 

For the purposes of the study, the clinical presentation at diagnosis was categorized as follows: malabsorption or anemia, gastrointestinal symptoms, extra-intestinal symptoms, or asymptomatic. Malabsorption was defined as weight loss and/or characteristic laboratory abnormalities, such as low folate or hypoalbuminemia. Gastrointestinal symptoms included abdominal pain, diarrhea, constipation, heartburn, and dysphagia. Extra-intestinal manifestations included dermatitis herpetiformis, recurrent aphthous stomatitis, enamel damage as confirmed by a dentist, failure to thrive (pediatric diagnosis), ataxia, unspecific arthritis, and elevated liver enzymes that were normalized by a gluten-free diet [19]. Non-specific and/or vague symptoms such as fatigue, infertility, back pain, and headache were disregarded. A patient could have had several symptoms simultaneously at diagnosis and thus be included in several symptom groups. 

The diagnostic delay, defined as the duration of possible symptoms before the diagnosis, was also recorded and further divided into ≤5 years and >5 years. 

### 2.3. Histology

The results of the histopathologic evaluation of the small-bowel mucosal biopsies were collected from patient records. According to our national guidelines, at least four representative biopsies are routinely taken from the duodenum in cases of suspected celiac disease [20]. Only correctly orientated cuttings are accepted for precise morphometric evaluation [21]. The diagnosis of celiac disease is based on the demonstration of either total, subtotal, or partial villous atrophy, comparable to the Marsh–Oberhuber classifications IIIa, IIIb, and IIIc, respectively [22]. In cases of dermatitis herpetiformis, the diagnosis is based on demonstration of granular IgA deposits in the papillary dermis by direct immunofluorescence examination in a skin biopsy [20]. 

### 2.4. Serology and Genetics

Information on possible earlier determined celiac disease autoantibodies was obtained from patient records. In addition, serum endomysial (EmA) and tissue transglutaminase antibodies (tTGab) were measured in all participants from the blood samples taken at the study visit. EmA was measured by the indirect immunofluorescence method as previously described [20] and titers 1:≥5 were considered positive. A commercial ELISA test (QUANTA Lite h-tTG IgA, INOVA Diagnostics, San Diego, CA, USA) was used to test tTGab, with a cut-off of >30.0 U/l for seropositivity according to the manufacturer’s instructions. In cases of IgA deficiency, the autoantibodies were determined by IgG class. 

Genotyping for celiac disease-associated HLA alleles was performed with the SSP^TM^ DQB1 low-resolution kit (Olerup SSP AB, Saltsjöbaden, Sweden) and/or tagging SNP approach [23]. Haplotypes were categorized into HLA DQ2 positives (DQ2.5/DQX or DQ2.2/DQ7), HLA DQ8 positives (DQ8/DQX), and both DQ2 and DQ8 negatives. 

### 2.5. Statistical Analyses

Statistical analyses were performed using SPSS Statistics for Windows (IBM Corp. Armonk, NY, USA) and STATA Statistical Software (StataCorp. LP, Lakeway Drive, TX, USA). Categorical variables were studied by McNemar and Chi-Squared tests and continuous variables were studied by the Wilcoxon signed rank test. A *p*-value < 0.05 was considered statistically significant.

### 2.6. Ethics

The study design and patient enrolment was accepted by the Ethics Committee of Pirkanmaa Hospital District. The study protocol follows the ethical guidelines of the Declaration of Helsinki. All participants gave written informed consent.

## 3. Results

### 3.1. Clinical Data

Twenty-four (12%) of the later diagnosed siblings were diagnosed in the present study. Twenty-six (13%) of the 200 patients were <18 years of age (range 5–17 years) at the time of the study and 37 (19%) of the 200 patients were <18 years of age (range 2–17 years) at diagnosis. Of the latter 37 subjects, 21 were first diagnosed and 16 later diagnosed siblings. Among the sibling pairs, there was only one dizygotic twin pair and no identical twins. The index patients were significantly younger at diagnosis and more often females compared to the later diagnosed siblings (Table 1). Gastrointestinal symptoms were the most common and equally distributed presentation was observed in both groups, but the indexes had malabsorption/anemia and extra-intestinal symptoms significantly more often (Table 1). Five index patients and 42 siblings were detected by screening; twenty were asymptomatic, all of them siblings. Seven of the asymptomatic patients were <18 years of age at diagnosis. The index subjects had more severe histological damage (less partial and more subtotal villous atrophy) and were more often seronegative at diagnosis, whereas the groups did not differ in terms of current age, length of diagnostic delay, or the presence of fractures, malignancies, and autoimmune comorbidities (Table 1). When looking at the whole study cohort, patients suffering from malabsorption or anemia had more severe villous atrophy compared to the asymptomatic subjects (total 33% vs. 28%, subtotal 50% vs. 22% and partial 17% vs. 50%, *p* = 0.012, respectively).

Among all subjects, the most common gastrointestinal symptom was diarrhea (43%) and the most common extra-intestinal symptom was dermatitis herpetiformis (17%). When comparing the symptom subgroups between the siblings, diarrhea (42% vs. 44%, *p* = 0.878), abdominal pain (45% vs. 34%, *p* = 0.117), and constipation (6% vs. 5%, *p* = 1.000) were equally presented among the indexes and siblings, as were oral symptoms (4% vs. 1%, *p* = 0.375) and failure to thrive (7% vs. 3%, *p* = 0.289). Dermatitis herpetiformis was more common among the index patients than among the siblings (25% vs. 9%, *p* = 0.002). None of the study subjects had ataxia, arthritis, or elevated liver enzymes.

Altogether, 21 pairs were concordant and 79 pairs were discordant for the celiac disease phenotype, as defined in the present study (Table 2). Gastrointestinal symptoms represented the most common concordant phenotype, and it was observed in 14 of the 21 pairs with concordant disease manifestation. Regarding partial concordance, in 28 pairs the index subject suffered simultaneous gastrointestinal symptoms and malabsorption/anemia and/or extra-intestinal symptoms, while the sibling had only gastrointestinal symptoms. Conversely, in nine pairs the sibling had both gastrointestinal and other symptoms, while the index suffered only gastrointestinal symptoms.

### 3.2. Genetics

Celiac disease-related HLA haplotype data was available for 66 pairs (Table 3). The remaining 34 pairs had one or more allele missing from the HLA-typing and were excluded from the haplotype comparisons. The most common haplotype among both the indexes and siblings was DQ2.5/DQX, followed by DQ2.5 homozygosity, and DQ2.5/DQ8. The other celiac disease-associated HLA haplotypes were present only in a small number of individuals. The HLA haplotype was equal in 46 (70%) out of the 66 pairs (Table 4). Of them, there were 17 pairs with a concordant clinical presentation, of whom eight pairs (47%) had the same haplotype (DQ2.5/DQX in six pairs). Of the remaining 49 pairs with a discordant clinical presentation, 38 (78%) had equal haplotype (DQ2.5/DQX in 19 pairs). This difference in the prevalence of the same haplotypes between pairs with and without concordant clinical presentation (47% vs. 78%) was statistically significant (*p* = 0.018).

## 4. Discussion

We observed substantial phenotypic variation between the first diagnosed indexes and the later diagnosed siblings with celiac disease. Gastrointestinal symptoms were frequently seen in both siblings, but they often co-existed with additional randomly distributed extra-intestinal manifestations, and within a significant portion of the pairs the clinical presentation was completely different. Familial phenotype concordance has not been previously studied using a similar approach, but some studies have shown that the intestinal form of celiac disease and dermatitis herpetiformis can occur within the same family [24,25]. Interestingly, variation in the clinical phenotype is not restricted to celiac disease, as similar heterogeneity has been reported, for example, in inflammatory bowel disease [26] and systemic lupus erythematosus [27]. 

The haplotypes were even more likely to be similar if the siblings were discordant for the clinical presentation, suggesting that HLA genotype does not predict the clinical outcome. Previously, Karell et al. investigated the distribution of HLA haplotypes in 110 sibling pairs with dermal and intestinal celiac disease and, in accord with us, found no significant association with clinical outcome [25]. Mustalahti et al. studied 28 asymptomatic and symptomatic sibling pairs [13], while Greco et al. studied 145 patients categorized into 16 separate phenotypes [14], and no phenotype–HLA haplotype associations were observed in either study. In contrast, there have been studies reporting the association of non-HLA variants with distinct celiac disease phenotypes [28,29,30]. For instance, certain genotypes of haptoglobin and CTLA4 have been associated with clinically mild or silent disease [29,30], whereas a particular interleukin-10 genotype seems to predispose to early-onset and histologically severe disease [28]. In any case, the role of both HLA and non-HLA risk variants seems to be at most modest, as supported by the few studies conducted using monozygous twins. Hervonen et al. investigated the co-occurrence of intestinal disease and dermatitis herpetiformis in six monozygous twin pairs: three pairs had the concordant phenotype and two pairs had the discordant phenotype [5]. Bardella et al. reported variance in both the clinical presentation and even the overall risk of developing the disease in five monozygous twin pairs [18]. In comparison, the age of onset and the disease risk can also vary between monozygous twins in children with type 1 diabetes [31]. Thus, the limited role of genetics and the additional effect of environmental factors as modifiers of the disease risk and phenotype seem to be common features in autoimmune diseases. 

The environmental factors involved in modulating the celiac disease phenotype and its development remain undetermined, as does how their effect is mediated. Hitherto, studies have focused on searching for possible modifiers of general celiac disease risk. For example, high amounts of gluten and gastrointestinal infections in infancy may increase the risk [32,33], whereas cesarean section, the age of gluten introduction, and breastfeeding are unlikely to play a role [34,35,36]. It remains unclear if these same factors affect the phenotype. One topic of interest relevant here is gut microbiota. We previously studied the duodenal microbiota of 32 celiac disease patients and observed that the bacterial profile, as well as the overall richness and diversity of the microbiota, varies depending on the phenotype [17]. The causality of these findings is difficult to evaluate, particularly since many of the previously mentioned environmental and genetic factors may affect both the structure and function of the microbiota [37,38]. In any case, a better understanding of the environmental factors could, besides further elucidating the pathogenesis, enable the development of interventions to reduce the disease risk and/or prevent the most severe outcomes. Large multicenter studies including patients with well-defined phenotypes are likely needed to fully decipher the complex association between phenotype, genotype, and environmental factors in celiac disease.

The first diagnosed siblings generally had a more severe disease presentation, demonstrated, for example, by their higher frequency of anemia and advanced villous atrophy. Similarly, Gudjónsdóttir et al. investigated the severity of symptoms in 105 sibling pairs and reported more severe symptoms among the indexes [29]. One explanation for the more advanced presentation in the first diagnosed siblings in the present study could be the longer disease history, even though the difference was not statistically significant [39]. Celiac disease is likely suspected with a lower threshold or screened even without apparent symptoms in the non-index siblings with a known family history for the condition, who can thus have a less severe phenotype despite being diagnosed at a later age. Another influencing factor could be the higher frequency of females among the indexes, as women have been shown to use more health care services compared to men [40]. In any case, the task of deciphering the factors behind differences in disease severity between siblings is complex, as various individual aspects may contribute to the timing of the diagnosis and the phenotype. In a broad sense, we found it important that the later diagnosed siblings had milder disease at diagnosis, as this indicates that family screening of celiac disease could simultaneously improve the diagnostic yield and prevent long-term complications due to earlier initiation of dietary treatment.

The main strength of the present study is the large and well-defined study cohort. The limitations are the subjective nature and challenging definition of the symptoms, plus possible recall bias. In order to categorize the phenotype as reliably as possible, the most non-specific symptoms were excluded, and one author made the classification and analyses systemically. However, the original diagnoses were made by several physicians, who could have had different approaches, for example, to clinical evaluation and laboratory testing. Altogether, complex interactions between the many confounders, such as the individual experience of symptoms, the implementation of screening, and differences between the groups in the ages at diagnosis and the number of subjects diagnosed in childhood, could influence the ultimate phenotype and were impossible to fully control for statistically. Genetic analysis was also limited to the assessment of the frequency of known celiac disease HLA risk haplotypes, and thus no deeper insight into the role of non-HLA genes and gene-to-gene interactions could be attained.

## 5. Conclusions

In conclusion, we found the clinical presentation of celiac disease to have a wide variation between the affected siblings, with the indexes generally having a more severe presentation at diagnosis. It is therefore important for physicians to remember possible atypical presentations, and to suspect the disease with a low threshold among the patient’s close relatives. Furthermore, HLA did not explain the differences, suggesting that non-HLA genes and environmental factors play significant roles. The ultimate reasons for the substantial phenotype variation in celiac disease remain to be determined in future studies.

## Figures and Tables

**Figure 1 nutrients-11-00479-f001:**
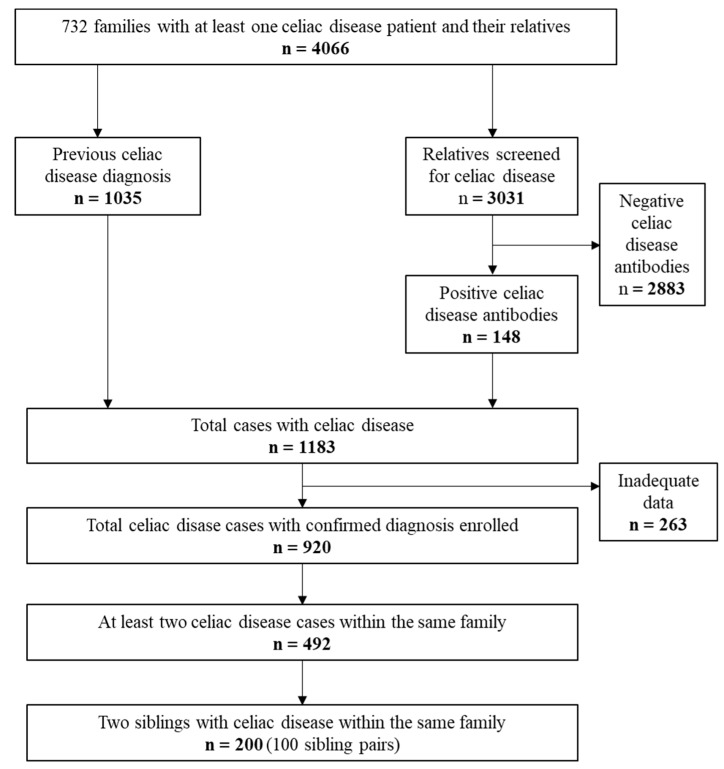
Flowchart of the study.

**Table 1 nutrients-11-00479-t001:** Diagnostic characteristics and presence of complications and comorbidities in 100 sibling pairs with celiac disease.

	Index Patients, *n* = 100	Siblings, *n* = 100	*p* Value
Age at diagnosis, median (Q_1_, Q_3_), year	37 (22, 47)	43 (25, 52)	<0.001
Age at study visit, median (Q_1_, Q_3_), year	52 (38, 59)	51 (36, 58)	0.704
Female, %	81	63	0.008
Diagnostic delay ^1^, %			0.073
>5 years	44	27	
≤5 years	56	74	
Clinical presentation at diagnosis, % ^2^			
Gastrointestinal	80	75	0.215
Malabsorption or anemia	45	20	<0.001
Extra-intestinal	33	12	<0.001
Asymptomatic	0	20	<0.001
Degree of villous atrophy at diagnosis, %			0.047
Total	29	27	
Subtotal	49	31	
Partial	20	41	
Positive celiac antibodies at diagnosis ^3^, %	89	100	0.030
Fractures, %	24	21	0.736
Malignancy ^4^, %	4	5	1.000
Associated diseases, %			
Thyroidal disease	20	9	0.052
Type 1 diabetes	1	4	0.375
Sjögren’s syndrome	2	1	1.000
IgA deficiency	1	0	1.000

^1^ Duration of symptoms before the diagnosis. Asymptomatic patients excluded. ^2^ Symptomatic patients could have had several overlapping presentations. ^3^ Tissue transglutaminase, endomysium, or reticulin antibodies. Data missing from 33 indexes and 23 siblings. Comparison made between 54 pairs. ^4^ For example, breast and thyroidal cancer.

**Table 2 nutrients-11-00479-t002:** Phenotype concordances at celiac disease diagnosis in 100 sibling pairs. Results are presented as numbers of sibling pairs.

Title		Index Patients, *n* = 100						
		GI	MA	EI	GI + EI	MA + EI	GI + MA	GI + MA + EI
Siblings, *n* = 100	GI	14	3	3	8	1	13	7
	MA	3	1	0	1	0	0	0
	EI	0	0	1	0	0	1	0
	GI + EI	3	0	2	1	0	3	0
	MA + EI	0	0	0	0	0	0	1
	GI + MA	6	1	1	1	1	4	0
	GI+ MA + EI	0	0	0	0	0	0	0
	Asymptomatic	6	4	2	3	0	5	0

GI, gastrointestinal; MA, malabsorption or anemia; EI, extra-intestinal. Pairs with concordant phenotype bolded.

**Table 3 nutrients-11-00479-t003:** Overall distribution of human leukocyte antigen (HLA) haplotypes in index cases and siblings with celiac disease. DQX = other than DQ2.5, DQ2.2, DQ7, or DQ8.

HLA	Index Patients, *n* = 66 %	Siblings, *n* = 66 %
*DQ2 positive*		
DQ2.5/DQX	48	53
DQ2.5/DQ2.5	29	21
DQ2.5/DQ8	11	12
DQ2.5/DQ2.2	2	5
DQ2.5/DQ7	3	2
DQ2.2/DQ7	0	2
*DQ2 negative, DQ8 positive*		
DQ8/DQX	2	2
DQ8/DQ8	0	0
DQ8/DQ2.2	2	0
DQ8/DQ7	3	3
*DQ2 negative, DQ8 negative*	2	3

**Table 4 nutrients-11-00479-t004:** Concordance of human leukocyte antigen (HLA) haplotypes in 66 sibling pairs. Results are presented as numbers of sibling pairs. Pairs with similar genotype bolded. DQX = other than DQ2.5, DQ2.2, DQ7, or DQ8.

		Index Patients, *n* = 66										
	HLA	DQ 2.5/X	DQ 2.5/2.5	DQ 2.5/8	DQ 2.5/2.2	DQ 2.5/7	DQ 2.2/7	DQ 8/X	DQ 8/8	DQ 8/2.2	DQ 8/7	DQ 2/8 Neg.
Siblings, *n* = 66	DQ 2.5/X	**25**	5	1	0	1	0	0	0	0	2	0
	DQ 2.5/2.5	2	**12**	0	0	0	0	0	0	0	0	0
	DQ 2.5/8	1	1	**6**	0	0	0	0	0	0	0	0
	DQ 2.5/2.2	2	0	0	**1**	0	0	0	0	0	0	0
	DQ 2.5/7	1	0	0	0	**0**	0	0	0	0	0	0
	DQ 2.2/7	0	0	0	0	1	**0**	0	0	0	0	0
	DQ 8/X	0	0	0	0	0	0	**1**	0	0	0	0
	DQ 8/8	0	0	0	0	0	0	0	**0**	0	0	0
	DQ 8/2.2	0	0	0	0	0	0	0	0	**0**	0	0
	DQ 8/7	0	1	0	0	0	0	0	0	1	**0**	0
	DQ2/8 Neg.	1	0	0	0	0	0	0	0	0	0	**1**

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
