# Peer review of "The Phenotype of Celiac Disease Has Low Concordance between Siblings, Despite a Similar Distribution of HLA Haplotypes"

_nutrients, 2019, doi:10.3390/nu11020479_

Round 1
Reviewer 1 Report
This is a study about clinical (presentation) concordance amongst adult siblings with celiac disease. The work is easy to read, well designed and perfectly exposed, including weaknesses. The results are inside what would be expected, but it is always interesting to support our empirical feelings with well processed data.
I only have one suggestion that perhaps might complete the information provided: to include, if possible, a table of haplotype concordance, similar to table 2 with clinical manifestations.
Author Response
Reviewer: I only have one suggestion that perhaps might complete the information provided: to include, if possible, a table of haplotype concordance, similar to table 2 with clinical manifestations.
Reply: We thank the Reviewer for this excellent suggestion and have now made a new Table 4. In order to present the table results we also made corresponding changes to the text. Please see Results, paragraph 3.2, lines 173-178 and Discussion, page 9, lines 192-193.
Reviewer 2 Report
Comments and suggestions
Study design and table 1.
1. Not clear how the N=492 cases were reduced to N=200. Explain what happen with the other cases.
2. Indicate the period of time from which the patients were selected e.g. Patients registered in the data base from 2000- 2018; has the criteria for diagnosis and registration changed during the period of time selected? Were the criteria for diagnosis of the pediatric population the same that for adults? Pediatric celiac population is often clinically viewed separate from adults and biopsy requirements can be different.
3. 13% of the 200 patients were under 18 years old. How they distributed and how young were they? The average age was 37 and 42, does this difference between the adult and pediatric population in your opinion influence the clinical symptomatology and the overall comparison expressed in Table 1. Explain and consider adding some comments on this regard.
Results:
1. May be important to emphasis that there were no statistical differences in the degree of villous atrophy between indexed and siblings. Is there any correlation between degree of villous atrophy and clinical manifestations such as malabsorption and the 20 cases asymptomatic? Make some considerations.
2. Also express your opinion on the number of newly diagnosed cases due to screening and the possible value of screening among siblings?
3. HLA haplotype table 3 indicates is available for 66 of 100 pairs. What happened to the remaining 34 pairs? Please elaborate.
Discussion
1. The discussion section needs more work to consider the above issues and a section discussing the limitations of the study and stronger points on the validity. Actually the abstract explains the conclusions better that the conclusion section. Consider rewriting.
Author Response
Reviewer 2.
Study design and table 1.
Reviewer: Not clear how the N=492 cases were reduced to N=200. Explain what happen with the other cases.
Reply: This issue is now further clarified; please see M&M, paragraph 2.1., page 2, lines 76-77.
Indicate the period of time from which the patients were selected e.g. Patients registered in the data base from 2000- 2018; has the criteria for diagnosis and registration changed during the period of time selected? Were the criteria for diagnosis of the pediatric population the same that for adults? Pediatric celiac population is often clinically viewed separate from adults and biopsy requirements can be different.
Reply: The adult and pediatric criteria were equal throughout the study as we have only very recently revised our diagnostic criteria according to ESPGHAN. We have now indicated these important issues as requested in M&M, paragraph 2.1, lines 65-67, M&M, paragraph 2.1, lines 70-71 and M&M, paragraph 2.1., page 2, lines 81-82.
13% of the 200 patients were under 18 years old. How they distributed and how young were they? The average age was 37 and 42, does this difference between the adult and pediatric population in your opinion influence the clinical symptomatology and the overall comparison expressed in Table 1. Explain and consider adding some comments on this regard.
Reply: Subjects diagnosed under 18 years of age were approximately equally distributed between the first and later diagnosed siblings, but there were slightly more children (21 vs 16) among the indexes which might have an influence on the symptoms. On the other hand, a greater part of the later diagnosed children (10/16 vs 2/21) was found by screening, illustrating again the complex interactions between these issues. This important issue has now been further clarified in Results, paragraph 3.1, lines 136-139 and lines 144-145 and discussed (page 10, lines 247-251).
Results
May be important to emphasis that there were no statistical differences in the degree of villous atrophy between indexed and siblings. Is there any correlation between degree of villous atrophy and clinical manifestations such as malabsorption and the 20 cases asymptomatic? Make some considerations.
Reply: In fact, although there was no difference in the prevalence of total villous atrophy, the index patients had significantly more often subtotal and less often partial villous atrophy than the siblings. This can be seen in Table 1 and is now further clarified in Results, paragraph 3.1, lines 145-148. In accord, subjects suffering from malabsorption and/or anemia had more severe villous atrophy compared to the asymptomatic siblings. This consideration can be now found in Results, paragraph 3.1, lines 148-151 and in Discussion, paragraph 4, lines 227-228.
Also express your opinion on the number of newly diagnosed cases due to screening and the possible value of screening among siblings?
Reply: This is indeed very important and we have now broadened the discussion regarding this issue as requested; please see Discussion, paragraph 4, lines 238-241.
HLA haplotype table 3 indicates is available for 66 of 100 pairs. What happened to the remaining 34 pairs? Please elaborate.
Reply: Unfortunately, there was no result available on both alleles from all of the sibling pairs, and thus a part of the pairs had to be excluded. This issue has now been elaborated in Results, paragraph 3.2, lines 169-171.
Discussion:
The discussion section needs more work to consider the above issues and a section discussing the limitations of the study and stronger points on the validity. Actually the abstract explains the conclusions better that the conclusion section. Consider rewriting.
Reply: We have now revised the Discussion based on the above comments by the Reviewer, and also harmonized the conclusion sections as requested.
Round 2
Reviewer 2 Report
Make sure to read a printed version of the document after deleting corrections and reformatting to ensure is ready for publication. Changes made are appropriated and improve the document.